# Electrodeposition of Bi-Te Thin Films on Silicon Wafer and Micro-Column Arrays on Microporous Glass Template

**DOI:** 10.3390/nano10030431

**Published:** 2020-02-28

**Authors:** Ning Su, Shuai Guo, Fu Li, Bo Li

**Affiliations:** 1International Graduate School of Shenzhen, Tsinghua University, Shenzhen 518065, China; 18842686901@163.com (N.S.); gs15@mails.tsinghua.edu.cn (S.G.); 2Shenzhen Key Laboratory of Advanced Thin Films and Applications, College of Physics and Optoelectronic Engineering, Shenzhen University, Shenzhen 518060, China

**Keywords:** Bi_2_Te_3_, electrodeposition, thermoelectric film, micro-column arrays, microporous glass template

## Abstract

Electrodeposition is an important method for preparing bismuth telluride (Bi_2_Te_3_)-based thermoelectric (TE) thin films and micro-column arrays. When the concentrations of Bi:Te in electrolytes were 3 mM:4 mM, the TE films satisfied the Bi_2_Te_3_ stoichiometry and had no dependence on deposition potential. With increasing over-potential, crystal grains changed from lamellar structures with uniform growth directions to large clusters with staggered dendrites, causing a decrease in the deposition density. Meanwhile, the preferred (110) orientation was diminished. The TE film deposited at −35 mV had an optimum conductivity of 2003.6 S/cm and a power factor of 2015.64 μW/mK^2^ at room temperature due to the (110)-preferred orientation. The electrodeposition of TE micro-columns in the template was recently used to fabricate high-power micro-thermoelectric generators (micro-TEG). Here, microporous glass templates were excellent templates for micro-TEG fabrication because of their low thermal conductivity, high insulation, and easy processing. A three-step pulsed-voltage deposition method was used for the fabrication of micro-columns with large aspect ratios, high filling rates, and high density. The resistance of a single TE micro-column with a 60 μm diameter and a 200 μm height was 6.22 Ω. This work laid the foundation for micro-TEG fabrication and improved performance.

## 1. Introduction

Thermoelectric (TE) devices perform the interconversion of electricity and heat based on the Seebeck, Peltier, and Thomson effects [1,2,3,4,5,6]. The conversion efficiency is relatively low when compared with other energy-conversion systems. However, because of their stability, with no transmission parts, no vibration and noise, and facile miniaturization, TE devices have been widely used for temperature difference power generation, refrigeration, and precise temperature control [7,8,9]. With the rapid development of micro-electromechanical systems (MEMS), there are important needs for managing micro-zone temperatures and powering passive devices. Therefore, TE miniaturization via MEMS has become common [10,11,12,13,14]. However, there are still many problems that limit their performance, such as the material properties, the reduced contact resistance between the materials and metal electrodes, and the preparation and arrangement of TE arm arrays with high-aspect ratios [15,16]. Therefore, while developing low-dimensional TE materials, optimizing the device structure and improving integration of the TE arms are keys to improving the micro-TE performance [17,18,19].

A typical TE device is composed of p- and n-type TE materials through thermal parallel and electrical series. In micro-TE devices, the basic elements are p- and n-type thin films or micro-column arrays. BiTe-based TE materials, such as n-type Bi_2_Te_3_ and p-type Sbi_2_Te_3_ have been proven to be the best around room temperature [11,20,21,22]. TE micro-devices prepared by their TE thin films and micro-columns array are widely used in micro-zone temperature-difference power generation at room temperature. Preparation of Bi_2_Te_3_ includes hydrothermal synthesis [23,24], molecular beam epitaxy [25,26], physical vapor deposition [27], magnetron sputtering [28,29,30], electrochemical deposition [11,12,31], and sintering [32,33]. Electrochemical deposition is widely used because of low cost, simple preparation, no special temperature or pressure requirements, and easy control of experimental parameters [11,34].

The fabrication of high-quality TE thin films and micro-column arrays is required for high-performance devices. There have been some works on fabricating high-performance n-type Bi_2_Te_3_ micro-column arrays [35,36,37,38,39]. Templates used for TE micro-device fabrication have included organic photoresists [35], anodic aluminum oxides (AAO) [37,38], silicon [36], and cut-glass [39]. Because of processing limits on the heights of micro-pores for the AAO and organic photoresist templates, the aspect ratios are relatively low, which results in smaller temperature differences and less output power. In addition, silicon and AAO templates have to be etched away after TE materials deposition because of their high thermal conductivity. This not only complicates the preparation process, but also severely impairs the mechanical properties of the device. Thus the template must have low thermal conductivity, high insulation, easy processing, and, for large-scale integration micro-TE devices, it must be amenable to dense TE micro-columns [40].

Here, Bi–Te thin films were initially fabricated on silicon wafers coated with Pt electrodes via three-electrode electrodeposition. The electrochemical behaviors of thin films deposited at different electrolyte potentials and different concentrations were investigated. The morphologies, elemental distributions, phase structures, and electrical properties of the thin films were characterized to determine the optimal electrolyte concentration and deposition potential for the subsequent deposition of TE micro-columns. For high-power micro-TEGs, TE micro-column arrays were fabricated with a microporous glass template that had low thermal conductivity and high insulation. The template had the following advantages. 1. Through-holes with high aspect ratios, alignment, and integration density were easily fabricated via laser etching. 2. It did not need to be etched after being filled with TE materials because of its high electric insulation and low thermal conductivity. In this way, the template supported the TE micro-columns and enhanced the mechanical properties of the entire device. 3. The low thermal conductivity ensured that the TE device could sustain a larger temperature difference during operation, which improved the output. A three-step pulsed deposition method was used to eliminate non-dense deposits due to concentration polarization. High-density TE micro-columns with high aspect ratios were fabricated on a highly integrated microporous glass template. The aspect ratio of the micro-columns could be increased by reducing the micro-pore diameter and increasing the glass template thickness. In this way, the micro-TEGs had larger TE arm integration and could support higher temperature differences.

## 2. Materials and Methods

Bi–Te thin films were prepared by three-electrode electrodeposition at room temperature. The TE films were deposited on silicon wafers using a constant voltage method. Figure 1a is a schematic of the three-electrode electrodeposition device. The working electrode (negative) was a square single-crystal silicon wafer with side length of 10 mm, which also sputtered with a layer of titanium (Ti) and platinum (Pt) electrodes (Ti/Pt = 20/150 nm, Beijing Bluebird Yuanxin Microsystem Technology Co., Ltd., Beijing, China). The silicon wafer was pre-cleaned in a solution of concentrated H_2_SO_4_ (98%) and H_2_O_2_ (30%) at a volume ratio of 2:1. The counter electrode (positive) was a square graphite sheet with a side length of 40 mm, which was electrochemically stable in an acidic solution. The area of the counter electrode was 16 times larger than that of the working electrode, which ensured a uniform distribution of the electric field between the two electrodes. The electrodes were connected to an electrochemical workstation (CHI-660E, CH Instruments, Shanghai, China) via clips. The reference electrode was saturated with potassium chloride calomel, which had a constant potential of 0.2412 V. Thus, the potentials discussed below were all relative to the reference electrode. Electrochemical deposition was performed in a beaker containing 200 mL of electrolyte that was an aqueous nitric acid solution of Bi_3_^+^ and HTeO_2_^+^ ions. The concentrated solution was formed by dissolving Te and Bi in concentrated nitric acid, followed by dilution with deionized water after all the elements were dissolved, and stored in a brown reagent bottle. The solutions were stable at room temperatures and could be stored for a long time. When using, the corresponding solutions were diluted with deionized water, and the pH was adjusted with dilute nitric acid to −0.5~0.5, according to the Bi–Te phase diagram. During electrodeposition, all three electrodes were immersed in the electrolyte. The working electrode was opposite the center position of the counter electrode, and the reference electrode was near the surface of the working electrode. With the output current of the electrochemical workstation, the following reduction reactions (Equations (1) and (2)) occurred at the cathode:Bi^3+^ + 3e^−^ = Bi(s),(1)
HTeO_2_^+^ + 3H^+^ + 4e^−^ = Te(s) + 2H_2_O,(2)

Te and Bi ions were reduced almost simultaneously because of similar reduction potentials. The electrochemical reaction can be expressed by Equation (3) [41]:2Bi^3+^ + 3HTeO_2_^+^ + 9H^+^ + 18e^−^ = Bi_2_Te_3_(s) + 6H_2_O,(3)

The electrochemical behaviors were investigated for the thin films deposited at different potentials in different electrolyte concentrations. Their morphologies, elemental distributions, and the phase and crystal structures were characterized with scanning electron microscopy (SEM; Zeiss SUPRA^®^ 55), energy-dispersive spectroscopy (EDS; Oxford X-Max 20) and X-ray diffraction (XRD; Rigaku, Japan). The electrical conductivities and Seebeck coefficients were measured simultaneously with a ZEM-3 (ULVAC-RIKO, Japan), and the power factors were calculated by PF = α^2^σ. The optimal electrolyte concentration and deposition potential were thus determined to maximize the performance of the deposited TE films.

Based on the thin-film depositions, electrodepositions of TE micro-columns on the microporous glass template were examined. The TE micro-column arrays were fabricated on the microporous glass templates via constant-voltage and pulsed-voltage depositions. The filling rates and densities of the TE micro-columns prepared by the two methods were analyzed via SEM, while EDS and XRD were used to characterize the elemental distributions and phases of the micro-column materials.

## 3. Results

### 3.1. TE Film

Figure 1b shows the cyclic voltammetry (CV) curves of the electrolyte with a 10 mM Te ion concentration and 7 mM, 7.5 mM, 8 mM, 9 mM, and 10 mM Bi ion concentrations. Figure 1c,d shows the CV curves of the unitary Bi (7.5 mM) and unitary Te (10 mM). The scanning speed of the electrochemical workstation was 5 mV/s. Every CV curve had one deposition peak corresponding to the synthesis of Bi–Te. The deposition potentials corresponding to the current peaks of all the Bi–Te alloys are on the positive side of that of the unitary Bi and Te, which is a typical phenomenon of under-potential deposition (UPD). During the negative scanning, the deposition reaction started when the potentials were lower than 0.2 V, and appeared as gray deposits on the silicon wafer surface. As the potentials became negative, the color of the deposits gradually darkened. The current peaks appeared when the potentials were −0.04 V. Hydrogen evolution occurred when the potentials were less than −0.4 V; many small bubbles were generated on the surface of the working electrode. Except for the electrolyte having a 10 mM Bi concentration, the electrolytes had one dissolution peak with a 0.48 V potential. The deposits on the silicon wafer gradually dissolved and the color of the surface became lighter with positive scanning. The potentials corresponding to the deposition and dissolution peaks were similar for the different solution concentrations. The current density and charge integral of the deposition peaks increased with Bi ion concentration, which indicated that it accelerated the deposition rate. The peak shapes in the CV curves were not symmetrical, and the large potential differences between the deposition and dissolution peaks indicated that this was not a reversible redox reaction.

Figure 2 shows the elemental composition of the deposits for different electrolyte concentrations and deposition potentials. The concentrations of Bi:Te in electrolytes were (a):7 mM:10 mM, (b):7.5 mM:10 mM, (c):8 mM:10 mM, (d):9 mM:10 mM, (e):10 mM:10 mM, and (f):15 mM:20 mM. The composition of the TE films depended on the electrolyte, but the elemental ratios of the deposits did not equal to those of the electrolyte. Furthermore, the composition of the deposits had no dependence on the over-potential. The deposits satisfied with the nominal stoichiometric ratio of Bi_2_Te_3_ when the Bi:Te concentrations in the electrolytes were 7.5 mM:10 mM and 15 mM:20 mM; that is 3:4 for the Bi:Te in the electrolyte. The composition of the deposits was less than 2:3 when the electrolyte concentration ratio was less than 3:4, because the HTeO_2_^+^ ions were preferentially adsorbed on the Pt substrate during electrodeposition. The electrodeposition rate had a strong relationship with the adsorption capacity of the HTeO_2_^+^ ions. When the HTeO_2_^+^ concentration in the electrolyte increased, the number of ions adsorbed on the Pt electrode surface increased, which increased the reaction rate [42]. The adsorption capacity of HTeO_2_^+^ ions was not related to the over-potential. The composition of the electrodeposited films can be expressed by Equation (4):X_Te_ = (i_Te_/Z_Te_)/{(i_Bi_/Z_Bi_) + (i_Te_/Z_Te_)},(4)
where X_Te_ was the Te elemental composition in the deposits, Z_Te_ and Z_Bi_ were the valences, and i_Te_ and i_Bi_ were the limiting current values during deposition. In the equation, the elemental composition in the deposits was only related to Z_Te_, Z_Bi_, i_Te_ and i_Bi_, i.e., it was only related to the concentration of the electrolytic solution and not the over-potential [43,44,45,46,47].

The compositions of the films were characterized when the Bi:Te concentrations in the electrolytes were 7.5 mM:10 mM and 15 mM:20 mM. The composition of the deposits did not change substantially when both ion concentrations were doubled. Figure 3a,c shows the deposit morphologies when deposited at −50 mV and −90 mV from the electrolyte containing 7.5 mM Bi and 10 mM Te. Similarly, Figure 3b,d shows the morphologies when deposited at −50 mV and −90 mV from the electrolyte containing 15 mM Bi and 20 mM Te. At −50 mV with a concentration of 7.5 mM Bi and 10 mM Te in the electrolyte, the deposit was fine lamellar grains with good crystallinity that were tightly packed with a uniform size distribution (Figure 3a). While at −50 mV with a concentration of 15 mM Bi and 20 mM Te in the electrolyte, the grains got bigger and had many branches. The branched grains were staggered and the size distribution was not uniform (Figure 3b). Smaller branches in the grains also appeared for −90 mV, in 7.5 mM Bi, and 10 mM Te (Figure 3c). For −90 mV, 15 mM Bi, and 20 mM Te, many small parallel branches grew from the previous branches, and the grains became thicker and looser (Figure 3d). This was because the equilibrium potential of the electrodeposition moved to the positive direction when the ion concentrations increased. Therefore, the over-potential was higher and the deposition rate was faster in the electrolytes with higher ion concentrations. The growth rate of the grains became much faster than the nucleation rate, resulting in coarse grains and branching [48,49]. Therefore, the film density deposited at high ion concentrations was less than that at low concentration.

Figure 3e shows the current density vs. deposition time at −50 mV for 7.5 mM Bi and 10 mM Te, and 15 mM Bi and 20 mM Te. At the beginning of the deposition, the current density instantly peaked at a very high value, then decreased rapidly, and then slowly decreased until it reached a stable current. Within 10 s from the beginning of electrodeposition, the current density increased from 44 mAcm^−2^ to 52 mAcm^−2^, then decreased to the initial value after 10 s. The rapid increase corresponded to rapid nucleation of Bi_3_^+^ and HTeO_2_^+^ ions on the surface of the Pt electrode. Within a few seconds, the surface was completely covered, with an increasing number of crystal nuclei. Next, the formation of new crystal nuclei occurred between the grains. The interface during electrodeposition was no longer electrode material, but newly generated TE material, which created a rapid increase in the electrolyte interface resistance. This was the root cause of the rapid decrease in current density. As the deposition proceeded, the electrolyte on the electrode surface formed a stable concentration polarization, and the concentration gradient gradually approached a stable value. The current density simultaneously reached a steady state that was higher for the high-concentration electrolyte, which indicated that the steady-state current density and the deposition rate could increase with increasing ion concentration.

Figure 4 shows the micro-morphology of the films deposited after 1200 s in the 7.5 mM Bi and 10 mM Te electrolyte. The deposition potentials were (a) −35 mV, (b) −60 mV, (c) −80 mV, (d) −120 mV, (e) −180 mV, and (f) −220 mV. The grains were lamellar with uniform sizes for the −35 mV and −60 mV potentials. The crystal grains were less than 1 μm in size with few branches. Each lamellar grain was closely arranged and grew perpendicular to the substrate. At −80 mV, the grains became coarse, the sizes were no longer uniform, and there were some quadrangular clusters. At −120 mV, more small dendrites parallel to each other were generated on the main crystal axis at a specific angle. The volume of the crystal grains and the gaps between the grains substantially increased. At −180 mV and −220 mV, the shuttle-shaped grains composed of lamellae were staggered in more directions to form larger clusters. These grains no longer grew perpendicular to the substrate, but uniformly in all directions. Overall, with increasing over-potential, the grain morphology changed from densely lamellar with uniform growth directions to large clusters with staggered dendrites. The preferred orientation disappeared and the gaps between grains increased, thus the density decreased. The color of the films turned from gray to black as the over-potential increased, and the surface roughness increased. Finally, the binding force of the deposits to the substrate decreased and they easily fell off [48,49,50].

The changing morphologies of the films with deposition potential were analyzed. The rate of crystal nucleation was larger for small over-potentials and was approximately equal to the growth rate of the crystal grain, resulting in uniform grain sizes. After a period of growth, the space in the direction parallel to the substrate was filled with new crystal nuclei. The grains were then pressed against each other, which suppressed their growth parallel to the substrate surface. Thus, the grains started to grow perpendicular to the substrate and eventually appeared lamellar. With increasing over-potential, the nucleation rate was reduced and became less than the growth rate. In addition, the current density increased, which produced heterogeneous grain growth. Some fast-growing grains were less affected by surrounding grains and grew not just perpendicular to the substrate, but in all directions. Hence, there was no preferred orientation of grain growth and more parallel dendrites were generated on the main crystal axis, leading to larger clusters and gaps between the grains [49,50,51].

Figure 5 shows the XRD diffraction patterns of the thin films obtained with different potentials in the electrolyte containing 7.5 mM Bi and 10 mM Te. The standard powder diffraction peak of Bi_2_Te_3_ (JCPDS # 15-0863) is shown at the bottom. The positions of the thin-film peaks for all the deposition potentials corresponded to the those of the standard, which indicated that the deposits were single-phase Bi_2_Te_3_ with no other Bi–Te compounds. However, the preferred (110) orientation appeared in the films deposited at a low over-potential. As the over-potential increased, the preferred orientation disappeared and the peak shape approached that of the standard [52]. This was consistent with the micro-morphology analysis.

Figure 6a,b shows the Bi and Te distributions in the thin films deposited at −35 mV in the electrolyte containing 7.5 mM Bi and 10 mM Te. The distributions were uniform and the Bi:Te ratio was 60.6:39.4, which was close to the Bi_2_Te_3_ stoichiometry. Figure 6c,d are micro-topographies of the oblique and cross sections of the corresponding film. It composes many 500-nm-diameter lamellar crystal grains with thicknesses of several tens of nanometers. The grains were close-packed perpendicular to the substrate surface. From the morphology and preferred orientation of the film, it could be inferred that the top of the lamellar grains (i.e., the part exposed on the upper surface of the film) corresponded to the (110) plane. Thus, the c-axis of the grains was parallel to the substrate. According to previous studies [53,54], Bi_2_Te_3_ is a semiconductor with a layered crystal structure, and its electrical conductivity is highly anisotropic. Studies have shown that the electrical conductivity of Bi_2_Te_3_ in the (001) plane is four times that in the c-axis direction, and that the Seebeck effect is isotropic. Therefore, for the film having a (110)-preferred orientation, the ZT value perpendicular to the substrate was approximately four times of that parallel to the substrate. The performance of in-plane TE devices could thus be improved by using the longitudinal direction of the film. In addition, differences in electrical conductivity can cause various electron transport difficulties during crystal growth. Electrons are less scattered and much easier to transport in the direction of high electrical conductivity. Therefore, the growth rates of crystal nuclei were larger along the direction perpendicular to the substrate, which would also increase the growth of film.

The TE thin film was deposited on a silicon wafer coated with a layer of Pt electrode. The Pt electrode had a higher electrical conductivity, and the silicon had a higher thermal conductivity. These factors affected the measurement of TE properties. To accurately measure the electrical properties, a cured epoxy resin was used to peel the film off the silicon wafer, so that it could be transferred to a substrate with lower thermal conductivity and higher insulation. Figure 7a,b shows the electrical conductivities, the Seebeck coefficients, and the power factors of thin films deposited at different potentials at room temperature. The electrical transport properties were analyzed by the measured carrier concentrations and mobilities at different deposition potentials (see Figure 7c). The Seebeck coefficients were negative, indicating that the deposits were n-type semiconductors. The Seebeck coefficients for films were deposited at different potentials changed little, ranging from −88.8 μV/K to −100.3 μV/K. The electrical conductivities, power factors, and carrier mobilities all decreased with increased over-potential. The carrier concentration decreased at first and then increased with increasing over-potential. The carrier concentration of the film deposited at −220 mV was much higher, probably because it contained more point defects. The electrical conductivity, power factor, and carrier mobility of the film deposited at −35 mV were 2003.6 S/cm, 2015.64 μW/mK^2^ and 52.50 cm^2^ V^−1^s^−1^, respectively, which were much higher than those films deposited at higher over-potentials. This was mainly because the (110) preferred orientation of the film enhanced its performance of electronic transmission. Figure 8a,b shows the Seebeck coefficients, electrical resistivities, and power factors of thin films deposited at −35 mV over the range of room temperature to 448.15 K. Both the absolute values of the Seebeck coefficient and the resistivities increased with increasing the temperature. The calculated power factors decreased as the temperature increased, with a maximum value of 2015.64 μW/mK^2^ at room temperature. Therefore, the film with a (110) preferred orientation had excellent electrical properties at room temperature, which optimized the electrodeposition of in-plane TE devices and TE micro columns.

### 3.2. TE Columns

High aspect ratio and high-integration TE micro-columns were deposited on the microporous glass template. Figure 9a,b shows physical and optical micrographs of a microporous glass template, respectively. The outer size of the glass template is 2 cm × 2 cm and the thickness is 200 μm. A series of regularly arranged through holes with a diameter of 60 μm and a spacing of 200 μm were created via laser etching at the center 1 cm × 1 cm area. The non-conductive glass template was fixed on the silicon wafer via patterned polyethylene glycol terephthalate tape during electrochemical deposition (see Figure 9c) and then connected to the negative electrode of the electrochemical workstation. Figure 9d shows the CV curves of the electrolyte with a 7.5 mM Bi ion concentration and 10 mM Te ion concentration when scanning on the silicon wafer and the microporous glass template. The current densities of the deposition and dissolution peaks on the glass template were much smaller than those on the silicon wafer. The −90 mV potential corresponding to the deposition peak on the glass was more negative than that for the silicon (−35 mV). Therefore, deposition on the microporous glass template was more difficult, and a larger driving force was required.

Constant-voltage deposition method was initially used to fabricate micro-columns in 7.5 mM Bi and 10 mM Te. Figure 10a,b shows the morphologies of the micro-columns deposited after 3 hours at −120 mV. Only some of the micro-pores were filled with deposits, and the micro-column growth rate in each micro-pore was uneven. The enlarged view of a single micro-pore (Figure 10b) revealed that the density of the deposit was very low, because it was difficult for the electrolyte to enter micro-pores having a large aspect ratio. On one hand, the electrode was at the bottom of the micro-pores because of the electric insulation of the inner glass wall, which made the electrodeposition difficult. On the other hand, the diffusion rate of the ions could not keep up with the consumption rate in the electrolyte at the bottom of the micro-pores during deposition; thus, a strong concentration polarization occurred, resulting in uneven composition and poor density. To solve these problems, pulsed-voltage deposition was used. Figure 10c shows the morphology of the micro-column array deposited over 15 h via two-step pulsed voltages. Specifically, the method involved a 4 s (t_on_) deposition at 100 mV (E_on_), followed by dissolution for 1 s (t_off_) at 500 mV (E_off_) to remove non-dense deposits and to allow the ions obtained by dissolution to enter the electrolyte. After cycling continuously for 15 h, the micro-pore filling rate had greatly increased, but 20% were still not filled with TE materials completely. Therefore, a three-step pulse voltage deposition method was used for complete filling of the templates. The first two steps were as described above. The third step involved a zero-current potential of 200 mV (E_0_) for 2 s (t_0_) to ensure sufficient ion diffusion in the electrolyte, which increased its consistency inside and outside the micro-pores. The three-pulse cycling was continuous until all the holes were filled, and then excess TE material was polished off the template surface. Figure 10d,e shows the morphology of the micro-column array deposited via Three-step pulsed voltages. The solid line in Figure 11a shows pulsed voltage vs. time during deposition, and the dashed line was current density vs. time. The current density rapidly increased to a maximum at the beginning of each deposition and then decreased until it reached a stable value. The increase was due to rapid nucleation, and the projected area of the crystal nuclei on the substrate also increased with the increasing nuclei. The current decreased when the number of crystal nuclei reached a certain value, and the concentration polarization in the electrolyte stabilized with the current. Figure 11b shows current vs. time for 100 cycles in the middle of the deposition. The peak deposition and dissolution currents were stable during the deposition until all the micro-pores were filled. The morphologies show a high density of filled micro-pores with few defects after a 15 h deposition and polishing. Hence, the three-step pulsed-voltage deposition improved the filling rate and density of the micro-columns [55,56,57,58,59].

Figure 12a,b shows the morphologies and elemental distributions at different locations in a single micro-column. The composition of the entire micro-column was close to the Bi_2_Te_3_ stoichiometry. It is also found that the Bi content increased slightly from the bottom to the top of the micro-column because of the different diffusion rates of Bi^3+^ and HTeO_2_^+^. The Bi^3+^ rate was lower than that of HTeO_2_^+^, so the Bi^3+^ concentration at the bottom of the micro-pores was slightly lower than that outside. In contrast, the HTeO_2_^+^ concentration was slightly higher at the bottom of the micro-pores than outside, which resulted in a lower Bi content at the bottom. While growing, the ratio of Bi and Te eventually became stoichiometric. When the micro-columns grew out of the micro-pores, the Bi content was slightly higher because there were more Bi^3+^ ions in the electrolyte. Hence, the elemental distribution of the entire micro-column was a gradient. Figure 12c shows the XRD pattern of filled TE micro-columns in a glass template. The diffraction peaks corresponded to the standard peaks of Bi_2_Te_3_, while the broad peak at a low angle corresponded to the amorphous glass template. The TE micro-columns had no preferred orientation. Figure 12d shows the current vs. voltage curve of a single micro-column via a two-probe method on the micro-probe platform. The data could be linearly fit, and the slope yielded the calculated 6.22 Ω resistance of the micro-column. The resistances of 20 micro-columns were measured, which shows an average resistivity of 6.6 × 10^−5^ Ωm. The resistivity of the TE micro-columns were comparable to that of bulk Bi_2_Te_3_ (10^−5^ Ωm) [33], considering the contact resistance between the probes and the deposits.

## 4. Conclusions

Three-electrode electrodeposition was used to prepare Bi–Te films on a Pt-coated silicon wafer. The composition of the TE films depended on the composition of the electrolyte, but not the over-potential. The steady-state current density and deposition rate increased with ion concentrations in the electrolyte. However, higher concentrations produced coarse grains and many more branches, which reduced the deposit density. As the over-potential increased, the crystal grains changed from lamellar structures with uniform growth directions to large clusters. At the same time, the (110)-preferred orientation disappeared. The electrical conductivities, power factors, and carrier mobilities of the n-type films decreased rapidly with increasing over-potential. The film deposited at −35 mV had good crystallinity due to the (110)-preferred orientation, which maximized the electrical conductivity at 2003.6 S/cm and the power factor at 2015.64 μW/mK^2^.

When a constant voltage method was used to fabricate TE micro-columns in the microporous glass template, the large aspect ratio of the micro-pores produced a strong concentration polarization, resulting in a low filling rate, low density, and an uneven deposit composition. A three-step pulse voltage deposition method was then used to prepare the micro-columns: E_on_ = −100 mV, t_on_ = 4 s; E_off_ = 500 mV, t_off_ = 1 s; E_0_ = 200 mV, t_0_ = 2 s. Continuously cycling for 15 h filled the holes with micro-columns with large aspect ratios and high densities. The composition of the entire micro-column was close to the Bi_2_Te_3_ stoichiometry, where the Bi content increased slightly from the bottom to top of the columns. The micro-columns had the Bi_2_Te_3_ crystal structure.

## Figures and Tables

**Figure 1 nanomaterials-10-00431-f001:**
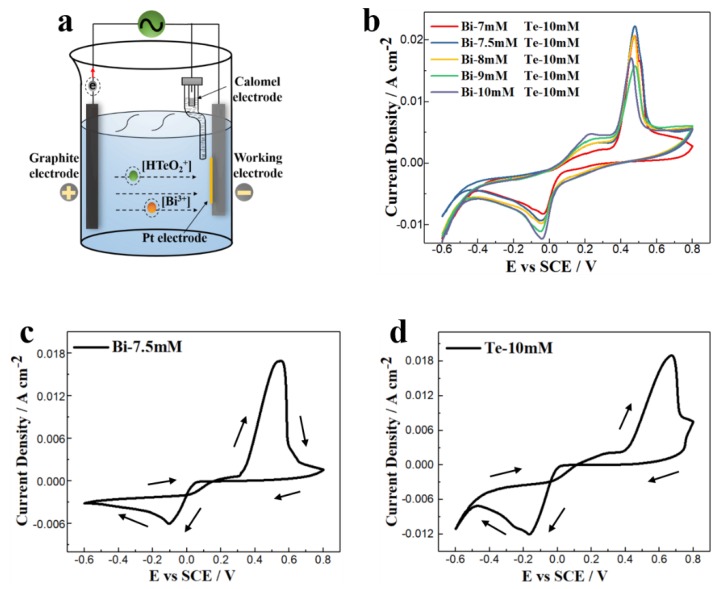
(**a**) The schematic of the three-electrode electrodeposition device. (**b**) Cyclic voltammetry curves of the electrolyte with a 10 mM Te concentration and 7 mM, 7.5 mM, 8 mM, 9 mM, and 10 mM Bi concentrations. (**c**,**d**) CV curves of the unitary Bi (7.5 mM) and unitary Te (10 mM).

**Figure 2 nanomaterials-10-00431-f002:**
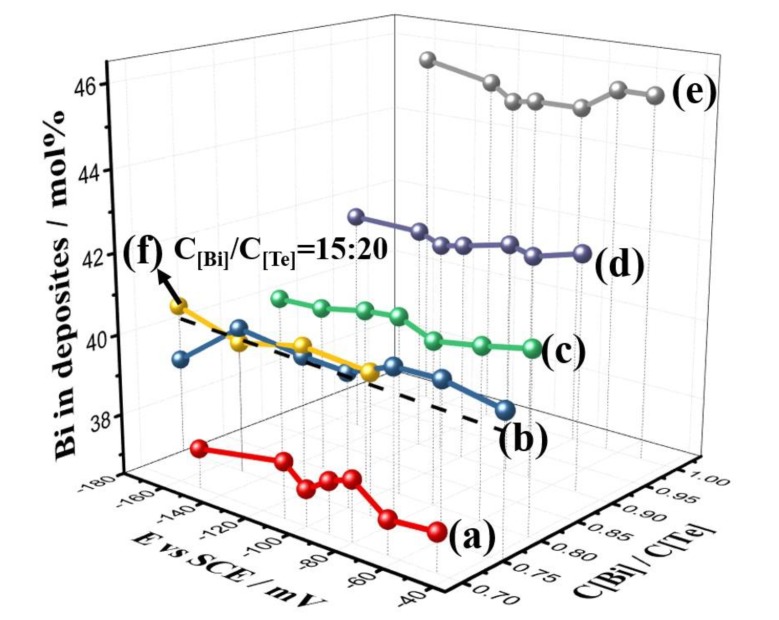
Elemental composition of the deposits for different electrolyte concentrations and deposition potentials.

**Figure 3 nanomaterials-10-00431-f003:**
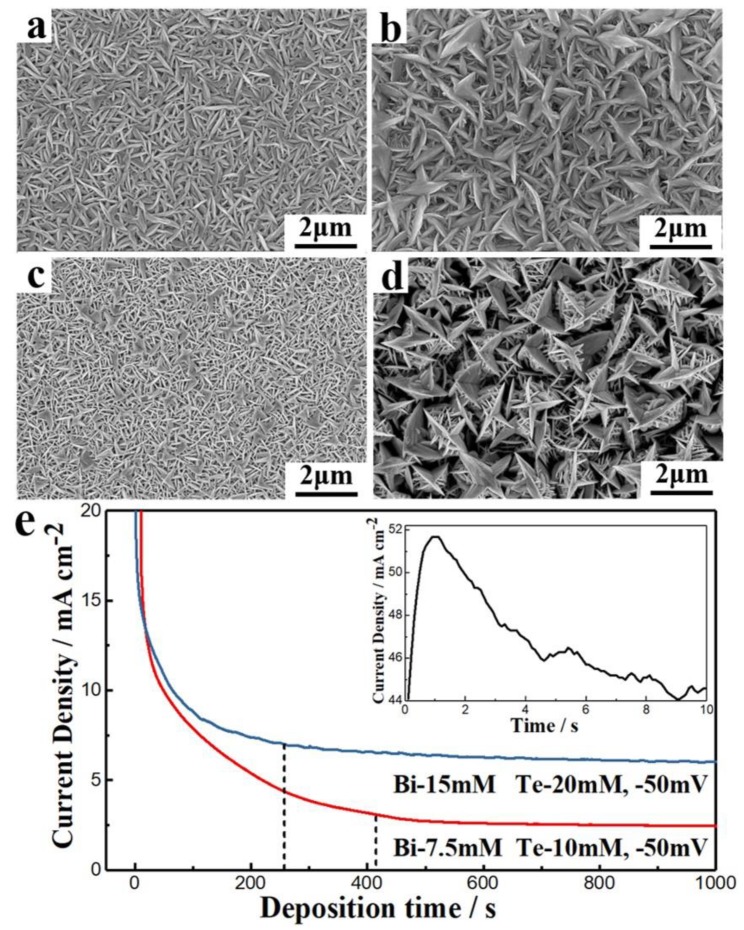
(**a**,**c**) The deposit morphologies when deposited at −50 mV and −90 mV from the electrolyte containing 7.5 mM Bi and 10 mM Te. (**b**,**d**) The morphologies when deposited at −50 mV and −90 mV from the electrolyte containing 15 mM Bi and 20 mM Te. (**e**) Current density vs. deposition time at −50 mV for 7.5 mM Bi and 10 mM Te, and 15 mM Bi and 20 mM Te.

**Figure 4 nanomaterials-10-00431-f004:**
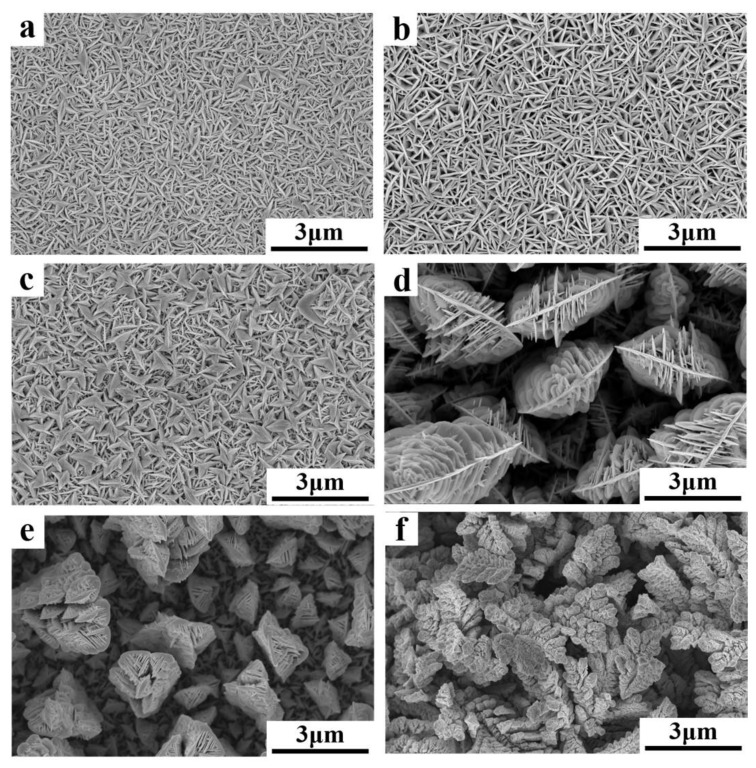
Micro-morphology of the films deposited after 1200 s in the 7.5 mM Bi and 10 mM Te electrolyte at the potential of (**a**) −35 mV, (**b**) −60 mV, (**c**) −80 mV, (**d**) −120 mV, (**e**) −180 mV, and (**f**) −220 mV.

**Figure 5 nanomaterials-10-00431-f005:**
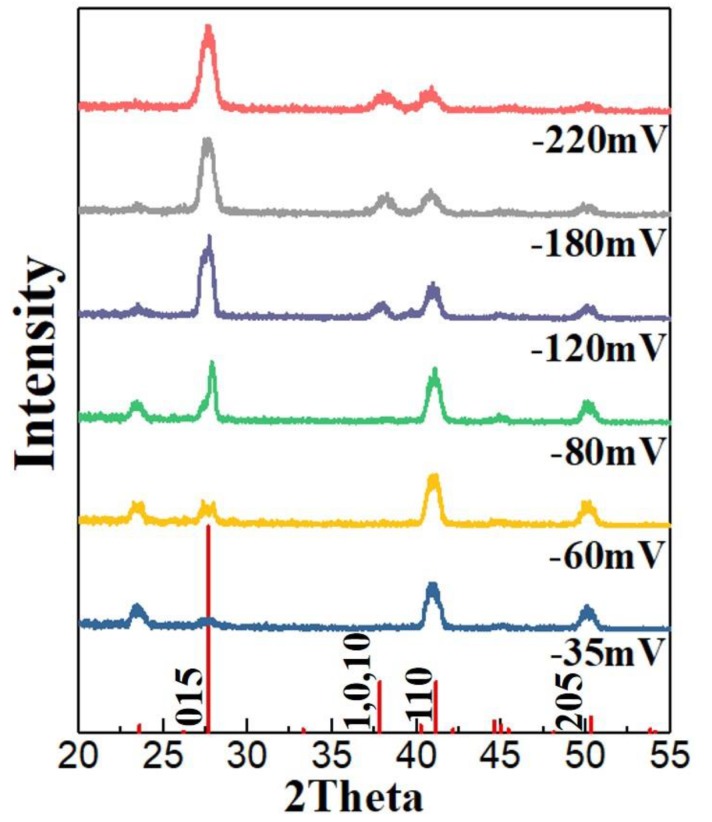
XRD diffraction patterns of the thin films obtained with different potentials in the electrolyte containing 7.5 mM Bi and 10 mM Te.

**Figure 6 nanomaterials-10-00431-f006:**
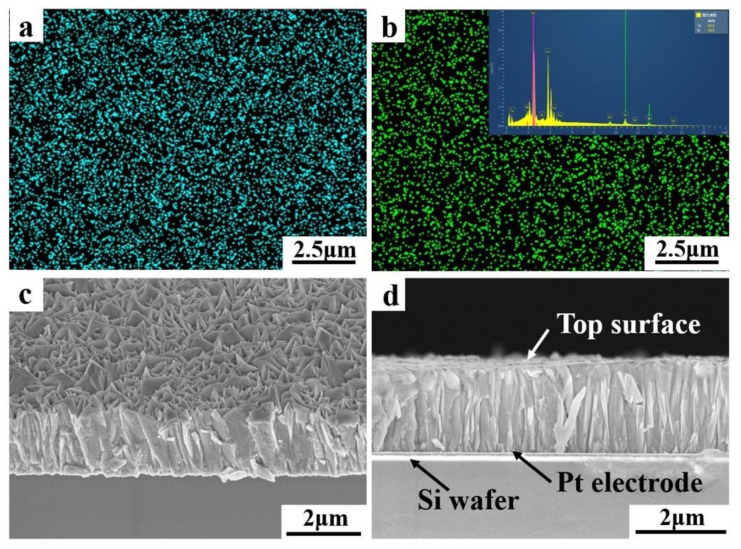
(**a**,**b**) Bi and Te distributions in the thin films deposited at −35 mV in the electrolyte containing 7.5 mM Bi and 10 mM Te. (**c**,**d**) Micro-topographies of the oblique and lengthwise sections of the film.

**Figure 7 nanomaterials-10-00431-f007:**
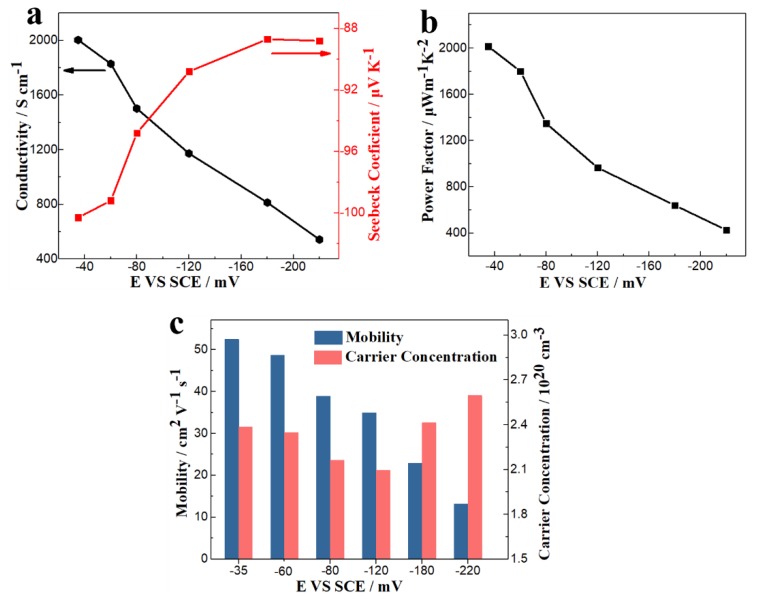
(**a**) Electrical conductivities and Seebeck coefficients, (**b**) power factors; (**c**) carrier concentrations and mobilities of thin films deposited at different potentials at room temperature.

**Figure 8 nanomaterials-10-00431-f008:**
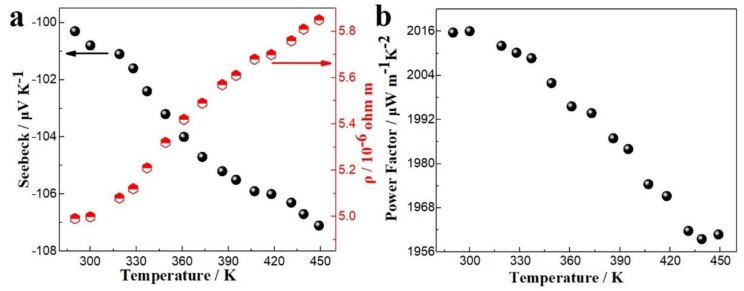
(**a**) The Seebeck coefficients, electrical resistivities and (**b**) power factors of thin films deposited at −35 mV over the range from room temperature to 448.15 K.

**Figure 9 nanomaterials-10-00431-f009:**
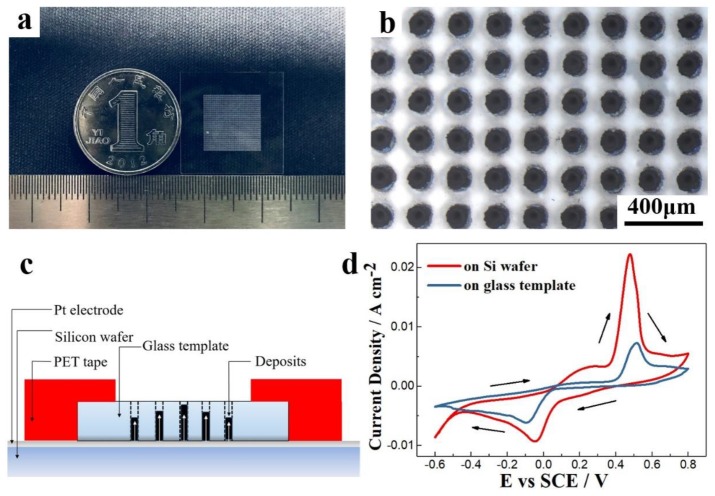
(**a**,**b**) Physical and optical micrographs of a microporous glass template. (**c**) The schematic of the electrode structure. (**d**) CV curves for 7.5 mM Bi and 10 mM Te when scanning on the silicon wafer and the microporous glass template.

**Figure 10 nanomaterials-10-00431-f010:**
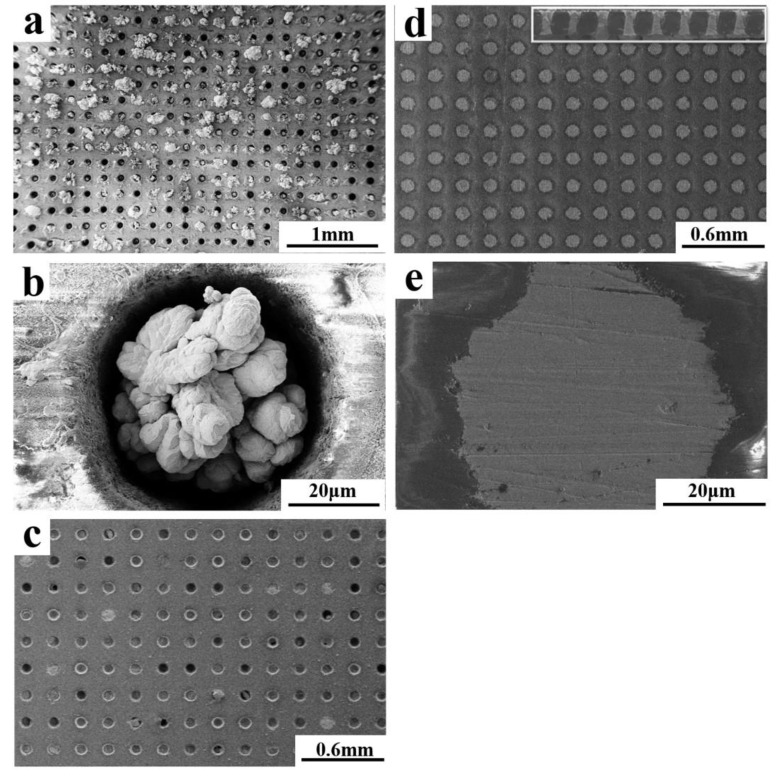
The morphologies of the micro-columns deposited over (**a**,**b**) 3 hours at −120 mV; (**c**) 15 h via two-step pulsed voltages; (**d**,**e**) 15 h via three-step pulsed voltages.

**Figure 11 nanomaterials-10-00431-f011:**
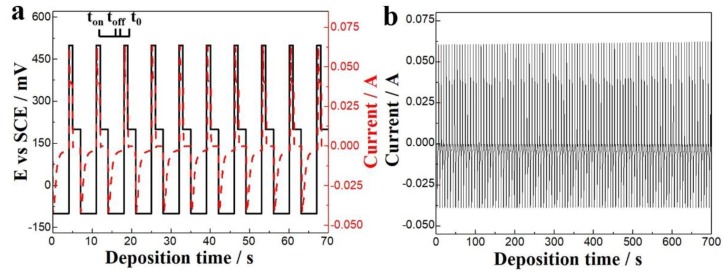
(**a**) The pulsed voltage and current vs. time during deposition. (**b**) The current vs. time for 100 cycles in the middle of the deposition.

**Figure 12 nanomaterials-10-00431-f012:**
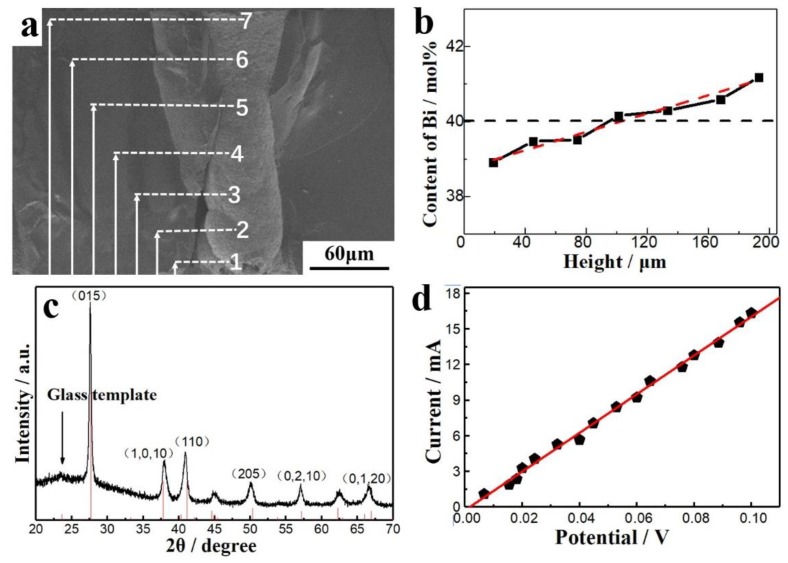
(**a**,**b**) The morphologies and elemental distributions at different locations in a single micro-column. (**c**) XRD pattern of the glass template filled with TE micro-columns. (**d**) The current vs. voltage curve of a single micro-column via a two-probes method on the micro-probe platform.

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
