# Peer review of "Electrodeposition of Bi-Te Thin Films on Silicon Wafer and Micro-Column Arrays on Microporous Glass Template"

_nanomaterials, 2020, doi:10.3390/nano10030431_

Round 1

Reviewer 1 Report

The authors show electrodeposition of BiTe alloy. Although, lot of literature is available on electrodeposition of BiTe alloys, the present study is of interest and is acceptable with few corrections.

CVs of individual Bi and Te must be shown and compared with Bi-Te alloy.

The argument in figure 5 must be substantiated with TEM and SAED. If not, remove that section.

In figure 8b, how was the different BiTe alloys were established that to with an accuracy of 3 decimal places?

Author Response

(1) CVs of individual Bi and Te must be shown and compared with Bi-Te alloy.

Reply: Thanks for the referee’s time and care in reviewing our manuscript. I have added the CVs of unitary Bi and Te in figure 1 in the revised manuscript. Besides, we have added the comparison of the CV curves between Bi/ Te and Bi-Te alloy in manuscript. Details are as follows:

Added: “Figures 1(c) and (d) show the CV curves of the unitary Bi (7.5mM) and unitary Te (10mM).”

“The deposition potentials corresponding to the current peaks of all the Bi-Te alloy is on the positive side of that of the unitary Bi and Te, which was a typical phenomenon of under-potential deposition (UPD).”

(2) The argument in figure 5 must be substantiated with TEM and SAED. If not, remove that section.

Reply: Thank you for pointing out the mistakes in the manuscript. I have removed the section about figure 5 and related description.

(3) In figure 8b, how was the different BiTe alloys were established that to with an accuracy of 3 decimal places?

Reply: The elemental distributions of thin films deposited at different electrolyte potentials and different concentrations were investigated in this study (shown in Figure 2). When the concentrations of Bi:Te in electrolytes were 7.5mM:10 mM, the TE films satisfied the Bi2Te3 stoichiometry and had no dependence on deposition potential. Then we tested the electrical properties of the thin films deposited at different potentials in the electrolyte containing 7.5 mM Bi and 10 mM Te (shown in Figure 7). We showed the element composition of the tested films in the form of BixTey in figure 7b. This expression may not be correct and we have removed it from the revised manuscript.  

Reviewer 2 Report

This work discusses the growth of Bi telluride by electrodeposition and the manufacturing of a device including columns. 

The process of electrodeposition is high quality, with a stoichiometry almost perfect. Unfortunately, the authors use concepts in an incorrect manner; for example, the reduction of ion and the crystallization cannot occur at the same time, as they claim, because first there must be a process of nucleation. 

On the other hand the morphology of the patterns are quite poor, making probably a low performance. The pattern that is being used to electroplate through patterns seems to be attached at the substrate; is there any attempt to avoid leaking between the holes from the bottom?

The authors may would like to use surfactants in order to grow higher quality patterns.

Finally, the authors do not provide the quality factor of the device. This is an important characteristics.

Overall, the work can be accepted, after minor changes as discussed above. 

Author Response

Reply:

(1) Thank you for pointing out the insufficient statements. I’m sorry that I made a mistake in some of the descriptions. We have corrected it in the revised manuscript. Details of the revision are as follows:

Original: “Every CV curve had one deposition peak corresponding to the synthesis of Bi-Te. Hence, the reduction of Bi3+and HTeO2+ ions and the crystallization of Bi-Te were completed simultaneously.”

Amended: “Every CV curve had one deposition peak corresponding to the synthesis of Bi-Te.”

(2) The morphologies shown in figure 10 are the upper surfaces of the micro-column arrays but not the surface close to the substrate. During deposition, the growth of each micro-column was affected by the surrounding micro-columns, and defects in the inner wall of micro-pore also affected the local current density and the growth rate in each micro-column, so the growing rate of every columns is usually different. Therefore, the deposition time was excessive to ensure that most of micro-pores can be filled up. At the end of the deposition, the upper surface of the template has many excess deposits. We will usually polish off the excess deposits to make a smoother upper surface of the micro-column arrays (such as figure 10 d and e).

(3) High-density TE micro-columns with high aspect ratios were fabricated on a highly integrated microporous glass template in this study. Then we fabricated TE micro-generators with excellent output performance based on the microporous glass template. The works of preparation and optimization of the TE micro-generators have been published in another study [1].

Reference

[1]    Su N, Guo S, Li F, et al. Micro-thermoelectric devices with large output power fabricated on a multi-channe glass template[J]. Journal of Micromechanics and Microengineering, 2018, 28(12).

Round 2

Reviewer 1 Report

The manuscript is acceptable